# Community Structure Diversity of Endophytic Fungi in *Cissampelos pareira* from Different Habitats and Their α-Glucosidase Inhibitory Activity

**DOI:** 10.3390/jof11090615

**Published:** 2025-08-22

**Authors:** Jing Yu, Cuiyun Yin, Zhaoyou Deng, Yin Yuan, Deying Tang, Xuanchao Shi, Yihang Li, Lixia Zhang

**Affiliations:** Yunnan Key Laboratory of Southern Medicine Utilization, Yunnan Branch Institute of Medicinal Plant Development, Chinese Academy of Medical Sciences, Jinghong 666100, China; yujingynu1113@163.com (J.Y.); Cyyin@implad.ac.cn (C.Y.); pharmacologyvip@163.com (Z.D.); swyuanyin@126.com (Y.Y.); tdy629@126.com (D.T.); xcs5977@163.com (X.S.)

**Keywords:** *Cissampelos pareira*, endophytic fungi, high-throughput sequencing, traditional culture methods, community composition, diviersity, α-glucosidase inhibitory activity

## Abstract

*Cissampelos pareira* is a medicinal plant with the potential effect of treating diabetes, commonly used by the Dai people in southern Yunnan Province. However, the wild resources of *C. pareira* are currently scarce. Endophytic fungi are a natural component of medicinal plants, while also serving as important repositories for discovering active natural products. In this study, we focused on 2-year-old *C. pareira* plants cultivated in potted and non-potted conditions. The community structure of endophytic fungi in the roots, stems, leaves, and flowers of two cultivation methods of *C. pareira* was investigated by using high-throughput sequencing (HTS) and traditional culture methods. Through HTS, we discover that the richness and diversity of endophytic fungi in *C. pareira* are associated with its growth environment and plant tissues. The endophytic fungi richness of *C. pareira* showed significant differences between the two habitats. And significant differences existed in the diversity of root endophytic fungi of *C. pareira* compared to those in the stems, leaves, and flowers. Additionally, the richness of endophytic fungi in the stems showed significant differences from that in the roots, leaves, and flowers. The results obtained using traditional culture methods revealed 69 endophytic fungi strains, classified into 2 phylum, 4 classes, 11 orders, 23 families, and 69 genera. The fermentation products of the obtained strains were evaluated for in vitro α-glucosidase inhibitory activity, and the results demonstrated that 11 endophytic fungi strains exhibited an inhibition rate exceeding 80%. The above-mentioned study can provide a theoretical basis for a comprehensive understanding of the community composition and diversity of endophytic fungi in *C. pareira*.

## 1. Introduction

Plant endophytic fungi are microbial communities that co-evolve with their host plants, living symbiotically within various plant organs and tissues without causing any apparent disease symptoms [1]. They are widely distributed across various plants, including algae, herbaceous plants, and woody plants. The abundance of endophytic fungi varies among host species; some harbor dozens of species, whereas others contain only a few [2]. Moreover, their diversity is influenced by factors such as the host plant’s habitat, growth season, and age [3,4,5]. Endophytic fungi and their host plants form a micro-ecosystem where the host plant provides nutrients and acts as a dispersal medium for the fungi. In return, these fungi can enhance the host plant’s resistance to various biotic and abiotic stresses, inducing oxidative stress responses in the plant to improve its adaptability and ecological competitiveness [6,7,8,9]. They can also produce plant hormones (such as indole acetic acid and gibberellins), siderophores, and increase the availability of inorganic elements to promote plant growth [10]. The importance of endophytic fungi for plant growth is self-evident, and studying the characteristics of plant endophytic fungi under different growth environments is particularly crucial. Current research on endophytic fungi in medicinal plants primarily focuses on the isolation, identification, and metabolic product analysis of host strains from a single tissue or a single environmental location [11,12]. To our knowledge, this is the first study to compare the community structure diversity of endophytic fungi between potted and non-potted conditions, representing two distinct habitats. These habitats can influence the growth of medicinal plants, highlighting the need for further investigation into the diversity of endophytic fungi across various tissues. Further studies on the secondary metabolites of endophytic fungi in medicinal plants have revealed that species diversity directly affects their ability to produce various structurally different metabolites, including compounds similar or identical to those found in their host plants [13,14]. Many of these metabolites exhibit a wide range of biological activities, such as anti-tumor, anti-malarial, anti-diabetic, antioxidant, and immunosuppressive properties. Consequently, endophytic fungi from medicinal plants have become an important resource for discovering new, effective, innovative, and low-toxicity natural bioactive lead compounds.

*Cissampelos pareira* is a perennial herbaceous plant belonging to the Menispermaceae family, primarily distributed in Asia, Africa, and the Americas. In Asia, it occurs in regions such as southern China, Indochina, Malaysia, Thailand, and India. This species has a long history of medicinal use in all regions where it is found. In the rainforests of South America, *C. pareira* is called “*Abuta*” and commonly known as the midwives’ herb. It is used to treat various women’s ailments, including menstrual cramps and uterine hemorrhage [15]. In traditional Indian medicine, *C. pareira* is known as “*Ambastha*” or “*Laghu patha*” and is commonly used to treat ailments such as malaria, heart diseases, cough, and fever [16,17]. At the same time, *C. pareira* is also a commonly used medicinal herb among the Dai people in Xishuangbanna, Yunnan, China, where it is known as “Yahunu”. It is believed to have properties that reduce swelling, alleviate pain, stop bleeding, and promote tissue regeneration [18]. Modern pharmacological studies have demonstrated that the *C. pareira* possesses various activities including antipyretic [19], analgesic [20], anti-diarrheal [21], antidiabetic [22], anticancer [23], anti-dengue virus [24], anti-fertility [25], anti-inflammatory [20], antioxidant [26], immunomodulatory [27], neuroprotective [28], and hepatoprotective effects [29]. Based on previous visits to traditional Dai doctors in Xishuangbanna by our research team, it was discovered that the Dai people use *C. pareira* as a hypoglycemic medicine. Similarly, in the traditional medical systems of India, the plant is incorporated into formulations for diabetes treatment [30]. It is also used for diabetes treatment in African countries like Kenya and in the American nation of Mexico [31,32].

Diabetes is a metabolic disease characterized by high blood sugar and is among the most prevalent non-communicable diseases worldwide [33]. It is generally classified into Type I and Type II, with Type I being an autoimmune disease and Type II being a preventable disease that accounts for approximately 90% of all cases [34]. Alpha-glucosidase is an enzyme that hydrolyzes carbohydrates into disaccharides and monosaccharides, which are subsequently absorbed by the small intestine and enter the bloodstream, making it a key factor in postprandial blood glucose levels [35]. Consequently, substances that can effectively inhibit α-glucosidase activity are considered valuable agents for alleviating postprandial hyperglycemia. Currently, only a limited number of α-glucosidase inhibitors (AGIs) are available for clinical use in diabetes management [36]. Recent studies have shown that, in addition to medicinal plants themselves, endophytic fungi isolated from these plants are emerging as important sources of novel AGIs [37]. Given the antidiabetic properties of *C. pareira* and the lack of research on its endophytic fungi, it is of great significance to study the diversity of endophytic fungi and their α-glucosidase inhibitory activity. Comprehensive information on the taxa and diversity of endophytic fungi of *C. pareira* is essential for the exploration and utilization of this microbial resource. Research on endophytic fungi diversity generally employs two technical approaches: culture-dependent methods and culture-independent methods. Although traditional culture methods are limited by subjectivity and yield only a subset of cultivable strains, they provide pure isolates for further functional and biochemical studies [38]. In contrast, culture-independent methods, such as HTS, are based on metagenomic techniques and directly analyze the DNA of fresh plant samples, thereby increasing the efficiency of microbial discovery. The potted and non-potted modes are two distinctly different habitats, and plant growth is closely related to the environment in which they are located. We hypothesize that the number and composition of endophytic fungi in plants may be related to the growth environment of the host plant. Additionally, the metabolites of endophytic fungi can largely exhibit biological activities similar to or the same as those of the host plant. In this study, to gain a more comprehensive understanding of the diversity of endophytic fungi in various tissues of *C. pareira* under different habitats, we employed both traditional culture methods and HTS to study potted and non-potted *C. pareira*. Additionally, we evaluated the α-glucosidase inhibitory activity of metabolites from the isolated and identified strains, providing a new approach for discovering active substances.

## 2. Materials and Methods

### 2.1. Collection and Pre-Processing of Samples

To compare the diversity of endophytic fungi in the roots, stems, leaves, and flowers of *C. pareira* under two habitat conditions, the research team transplanted cultivated seedling of *C. pareira* plants into field plots (non-potted) and experimental pots (potted, the soil used was consistent with that in the experimental field) at the South Medicine Garden in Jinghong City, Xishuangbanna, Yunnan Province (22°00′78″ N, 100°79′63″ E) in January 2022. The experimental soil was red soil (collected from the Southern Medicinal Garden, with a pH value of 4.7 and clay content of 34% as determined by the specific gravity method). The *C. pareira* in both habitats grew under semi-shaded conditions, with irregular watering based on specific weather conditions. In March 2024 (during the peak flowering period of *C. pareira*), healthy potted and non-potted *C. pareira* plants were collected (Appendix A). Three plants were collected from each growth condition, and the roots, stems, leaves, and flowers were pre-treated within 12 h as follows: each tissue was washed thoroughly with running water, and then in a super-clean workbench, the roots and stems were soaked in 3% sodium hypochlorite for 1 min, rinsed with sterile water three times, soaked in 75% ethanol for 1 min, and rinsed with sterile water three times. The leaves and flowers were only soaked in 75% ethanol for 1 min and rinsed with sterile water three times. Then, sterile filter paper was used to absorb the surface moisture from all tissues [39]. The final rinse was cultured in a Petri dish at 28 °C for 7 days to confirm complete sterilization. Each tissue was labeled (roots, stems, leaves, and flowers of potted *C. pareira* were labeled PR, PS, PL, and PF, respectively; roots, stems, leaves, and flowers of non-potted *C. pareira* were labeled FPR, FPS, FPL, and FPF, respectively), placed in a sterile EP tube, and stored in a −80 °C freezer.

### 2.2. DNA Extraction, and HTS

Total genomic DNA was extracted from *C. pareira* samples using the TGuide S96 Magnetic Soil/Stool DNA Kit (Tiangen Biotech (Beijing) Co., Ltd., Beijing, China) according to the manufacturer’s instructions. The quality and quantity of the extracted DNA were examined using electrophoresis on a 1.8% agarose gel, and DNA concentration and purity were determined with a NanoDrop 2000 UV-Vis spectrophotometer (Thermo Scientific, Wilmington, DE, USA). The hypervariable region ITS1-2 of the fungi ITS gene was amplified with primer pairs ITS1F: 5′-CTTGGTCATTTAGAGGAAGTAA-3′ and ITS2R: 5′-GCTGCGTTCTTCATCGATGC-3′ [40]. Both the forward and reverse ITS primers were tailed with sample-specific Illumina index sequences to allow for deep sequencing. The PCR was performed in a total reaction volume of 10 μL: DNA template 5–50 ng, forward primer (10 μM) 0.3 μL, reverse primer (10 μM) 0.3 μL, KOD FX Neo Buffer 5 μL, dNTP (2 mM each) 2 μL, KOD FX Neo 0.2 μL, and finally ddH_2_O up to 20 μL, after initial denaturation at 95 °C for 5 min, followed by 20 cycles of denaturation at 95 °C for 30 s, annealing at 50 °C for 30 s, and extension at 72 °C for 40 s, and a final step at 72 °C for 7 min. The amplified products were purified with the Omega DNA purification kit (E.Z.N.A.® Cycle Pure Kit D6492-02) (Omega Inc., Norcross, GA, USA) and then sent to Qingke Biotechnology (Qingdao, China) Co., Ltd. for library construction and sequencing. The sequencing data generated from the Illumina platform have been deposited in the NCBI Sequence Read Archive (SRA) under the accession number PRJNA1280751.

### 2.3. Analysis and Processing the Date of HTS

Trimmomatic (version 0.33), Cutadapt (version 1.9.1), and USEARCH (version 10) were used for quality control of the HTS data, and non-specific amplified sequences and chimeras were filtered to obtain valid sequences. Sequences with more than 97% similarity were clustered into the same Operational Taxonomic Units (OTUs) using USEARCH. Species annotation analysis (with a confidence threshold of 70%) was performed on the feature sequences based on the plain Bayesian classifier in QIIME2 (version 2020.6) using Unite (version 7.2) as the reference database, and then at each level (phylum, class, order, family, genus, species) the community composition of each sample was counted. The alpha diversity (Simpson index and Chao1 index) were calculated and displayed by the QIIME2 and R software (version 3.3.1), respectively. Beta diversity analysis was performed using QIIME2-software to compare the degree of similarity that exists among different samples in terms of species diversity, such as Principal Coordinate Analysis (PCoA) and unweighted group average method (UPMGA) to analyze the relationship between the fungi community structure in different samples.

### 2.4. Isolation and Identification of Endophytic Fungi Based on Culture Medium Methods

The collected tissues of *C. pareira* were washed thoroughly with running water. In the sterile workbench, the samples were surface sterilized sequentially with 3% sodium hypochlorite, sterile water, 75% ethanol, and sterile water (as detailed in Section 2.1). The samples were then cut into small pieces and evenly placed in Petri dishes containing PDA medium supplemented with 150 mg/L of penicillin and kanamycin. Each Petri dish contained 5 tissue pieces, and they were incubated at 28 °C for 7–10 days. Observations were made daily, and the edge mycelia of colonies were picked using a sterile inoculation needle and subcultured onto new Petri dishes. This process was repeated 2–3 times to purify the strain until a single colony was obtained. The purified strains were inoculated onto slant culture media and stored in a refrigerator at 4 °C at the Yunnan Branch Institute of Medicinal Plant Development, Chinese Academy of Medical Sciences.

Using a fungi genomic DNA rapid extraction kit, total DNA from mycelia was extracted and amplified with primers ITS1 (5′-TCCGTAGGTGAACCTGCGG-3′) and ITS4 (5′-TCCTCCGCTTATTGATATGC-3′) [41]. The PCR reaction system (25 μL) included: 12.5 μL of Premix Taq, 0.5 μL of ITS1 (10 μmol/L), 0.5 μL of ITS4 (10 μmol/L), 1 μL of DNA template (20 ng/μL), and 10.5 μL of RNase-free water. The PCR reaction conditions were: 94 °C for 5 min; 94 °C for 60 s, 53 °C for 60 s, 72 °C for 2 min, for 30 cycles; and a final extension at 72 °C for 10 min. The PCR products were checked by 1% agarose gel electrophoresis to confirm clear product bands and then sent to Qingke Biotechnology Co., Ltd. (Kunming) for sequencing. The obtained rDNA ITS sequences of endophytic fungi were searched in the NCBI database using BLAST+ (version 2.15.0) (https://blast.ncbi.nlm.nih.gov/Blast.cgi) accessed on 5 November 2024, and the species with the highest similarity to the uploaded sequences were recorded, and the phylogenetic tree of the strains was constructed based on Kimura 2-parameter distances and detected by the bootstrap method with 1000 times of self-expansion, using neighbor-joining methods (NJ) in MEGA 7.0. The obtained ITS sequences were deposited in the NCBI GenBank database with assigned accession numbers.

### 2.5. Alpha-Glucosidase Inhibitory Activity of Endophytic Fungi Fermentation Products

The purified endophytic fungi were inoculated into Potato Dextrose Broth (PDB) medium and incubated on a constant-temperature shaking incubator for 6–8 days (temperature 28 °C, shaking speed 120 r/min). After incubation, an equal volume of ethyl acetate was added for ultrasonic extraction for 30 min, repeated three times. The combined extracts were concentrated by rotary evaporation to obtain the extract, which is the ethyl acetate crude extract of the fungal fermentation liquid (ethyl acetate extract, EA). This sample is the test sample. The test sample and the positive control, acarbose, were dissolved in 0.1 mol/L phosphate-buffered solution (pH = 6.8) to prepare a 0.5 mg/mL stock solution. The α-glucosidase activity was measured in vitro using the improved method by Yang et al. [42]. The experiment was performed in a 96-well plate, with a total reaction volume of 200 μL. First, 50 μL of phosphate-buffered solution (0.1 mol/L, pH = 6.8) was added, followed by 10 μL of α-glucosidase (0.2 U/mL), and finally, 20 μL of the test sample solution was added. The mixture was incubated at 37 °C for 15 min. Then, 20 μL of p-Nitrophenyl α-D-glucopyranoside (PNPG) (2.5 mM) was added, and the reaction continued at 37 °C for 15 min. Afterward, 100 μL of sodium carbonate solution (2.5 mM) was added to stop the reaction. The absorbance (A) value was measured at a wavelength of 405 nm after mixing thoroughly. The α-glucosidase inhibition rate was calculated using the formula: I (%) = [1 − (A (Sample) − A (Blank))/A (Control)] × 100%. Where A (Sample) is the absorbance of the sample; A (Blank) is the absorbance of the control group without the sample (negative control); A (Control) is the absorbance of the blank group without α-glucosidase. All operations were repeated three times, and the experimental data are expressed as the mean ± standard deviation.

### 2.6. Statistical Analysis

This study comprised two distinct treatments (involving four plant tissues), with a total of 24 biological samples collected. The data exhibited normal distribution (Shapiro–Wilk test, *p* > 0.05), thus parametric tests were employed. Rarefaction curves for potted cultivated *C. pareira* samples reached saturation at 5000 sequences per sample, while those for non-potted cultivated *C. pareira* plateaued at 10,000 sequences per sample, demonstrating adequate sequencing depth. Statistical analysis of endophytic fungi richness (Chao1 index) and diversity (Simpson index) in root, stem, leaf, and flower tissues from both potted cultivated and non-potted cultivated *C. pareira* was performed using one-way ANOVA in SPSS 22.0. Inter-group differences were analyzed with Tukey’s HSD test (*p* < 0.05). Principal coordinate analysis (PCoA) and cluster analysis (Bray–Curtis distance) were conducted based on ANOSIM to examine fungal communities across different tissues in the two habitats.

## 3. Results

### 3.1. Analysis of the Endophytic Fungi Community Structure of C. pareira Based on the HTS Method

#### 3.1.1. Sequencing Results and Alpha Diversity Analysis

After processing the raw data obtained from 24 samples (with three biological replicates per tissue) of roots, stems, leaves, and flowers from both potted and non-potted *C. pareira*, 1,534,274 valid sequences were ultimately obtained, with an average sequence length of 305 bp. Specifically, the average sequence lengths for potted *C. pareira* were 291, 336, 317, and 292 bp in roots, stems, leaves, and flowers, respectively, while those for non-potted *C. pareira* were 340, 281, 275, and 310 bp in the corresponding tissues. The dilution curves of all samples showed gradual plateauing as sequencing depth increased (Appendix A), and the OTU coverage indices for all tissue types exceeded 0.99 (Appendix A). These results demonstrate the rationality of the sequencing data and indicate that the findings adequately reflect the true microbial diversity across different tissues of *C. pareira*.

To comprehensively assess the alpha diversity of microbial communities, this experiment utilized the Simpson index to characterize the diversity of endophytic fungi and the Chao1 index to evaluate their richness. As shown in Figure 1, the alpha diversity indices of endophytic fungi in various tissues of potted and non-potted *C. pareira* were analyzed. Analysis of the Simpson index revealed that the stems of non-potted *C. pareira* exhibited the highest endophytic fungi diversity (index = 0.9716), followed by leaves (0.9685) and flowers (0.9616), while roots showed the lowest diversity (0.4299). These results indicate statistically significant differences in endophytic fungi diversity between stems and roots (*p* = 0.043), leaves and roots (*p* = 0.043), and flowers and roots (*p* = 0.044). In contrast, potted *C. pareira* displayed minimal differences in fungi diversity across roots, stems, leaves, and flowers with the indices of 0.8459, 0.6814, 0.5802, and 0.8792 (*p* > 0.05) (Figure 1a and Appendix A). Furthermore, analysis of the Chao1 index (Figure 1b and Appendix A) demonstrated that the stems of non-potted *C. pareira* exhibited the highest endophytic fungi richness (1618.4540), which was significantly greater than that of roots (618.9688, *p* = 0.014), leaves (909.3271, *p* = 0.0065), and flowers (743.0725, *p* = 0.0082). However, the richness of endophytic fungi in the four tissue parts (root, stem, flower, and leaf) of potted *C. pareira* had little difference with the indices of 386.3586, 204.8444, 268.2485, 442.5682 (*p* > 0.05). Furthermore, the non-potted *C. pareira* exhibited Simpson and Chao1 indices of 0.8329 and 972.4556, respectively, compared to 0.7467 and 325.5050 in potted *C. pareira* (Appendix A). Notably, the Chao1 index showed significant variation (*p* = 0.031), while the Simpson index demonstrated less pronounced differences (*p* = 0.59). Therefore, the richness of *C. pareira* endophytic fungi varied considerably due to differences in growth environment and tissue localization.

OTU clustering analysis yielded a total of 9587 OTUs (Appendix A). After excluding unclassified OTUs, these were classified into 18 phyla, 60 classes, 139 orders, 324 families, 851 genera, and 1607 species. As shown in Figure 2a, the unique OTUs in the eight tissues from different habitats were 458, 785, 466, 830, 2214, 763, 2004, and 1195, respectively, with only 6 shared OTUs. A total of 3129 OTUs were obtained from potted *C. pareira* (Appendix A). The unique OTUs in each tissue were 858, 531, 507, and 940, with 10 common genera. Shared OTUs between tissues were as follows: PS-PL (18), PR-PS (20), PS-PF (29), PR-PL (114), PF-PL (154), and PR-PF (158). Notably, the stem (PS) exhibited fewer shared OTUs with leaves (PL), roots (PR), and flowers (PF), suggesting distinct endophytic fungi community richness and diversity between stems and other tissues (Figure 2b). For non-potted *C. pareira*, a total of 6901 OTUs were obtained (Appendix A), with unique OTUs in each tissue being 782, 2394, 2070, and 1239, and 25 common genera. Shared OTUs between tissues were as follows: FPR-FPS (50), FPS-FPF (79), FPS-FPL (102), FPR-FPF (123), FPR-FPL (155), and FPL-FPF (208). The stem showed fewer shared OTUs with the root and flower, suggesting differences in the richness and diversity of endophytic fungi communities between the stem and these two tissues, as shown in Figure 2c.

The “shared OTUs” refer to OTUs that appear in different samples. Among the 8 samples of potted and non-potted *C. pareira* (Appendix A), there are 6 shared OTUs: OTU14, OTU21, OTU39, OTU40, OTU239, and OTU739. In the 4 different tissues of potted *C. pareira*, there are 10 OTUs: OTU14, OTU21, OTU39, OTU40, OTU121, OTU239, OTU656, OTU739, OTU1097, and OTU4616. In the 4 different tissues of non-potted *C. pareira*, there are 25 OTUs: OTU5, OTU14, OTU15, OTU21, OTU38, OTU39, OTU40, OTU41, OTU69, OTU71, OTU78, OTU86, OTU102, OTU125, OTU133, OTU159, OTU162, OTU169, OTU174, OTU239, OTU255, OTU461, OTU559, OTU739, and OTU818. Among them, OTU14, OTU21, OTU39, OTU40, and OTU239 appear in *C. pareira* from both environments, which may indicate that these OTUs represent microorganisms that can survive in different environments or conditions, or they have strong environmental adaptability.

“Unique OTUs” refer to OTUs that are specific to certain tissues, meaning they are not found in other groups. For example, in the root tissue of potted *C. pareira* (Appendix A), unique OTUs include OTU17, OTU290, OTU310, etc.; in the stem tissue, unique OTUs include OTU25, OTU29, OTU36, etc.; in the leaf tissue, unique OTUs include OTU76, OTU95, OTU199, etc.; and in the flower tissue, unique OTUs include OTU52, OTU53, OTU116, etc. In the root tissue of non-potted *C. pareira*, unique OTUs include OTU3938, OTU3938, OTU3940, etc.; in the stem tissue, unique OTUs include OTU4879, OTU4880, OTU4881, etc.; in the leaf tissue, unique OTUs include OTU1554, OTU1555, OTU1556, etc.; and in the flower tissue, unique OTUs include OTU9, OTU18, OTU19, etc. These unique OTUs may be influenced by specific environmental factors, treatment conditions, or sample characteristics. Through the analysis of shared and unique OTUs, we can gain a better understanding of the microbial community composition and its ecological functions under different environments or conditions.

#### 3.1.2. Beta Diversity Index

Beta diversity of endophytic fungi communities in different tissues of potted and non-potted *C. pareira* was analyzed using Principal Coordinate Analysis (PCoA) based on Bray–Curtis distance and cluster analysis. The PCoA results at the OTU level are shown in Figure 3a. The first principal coordinate (PCo1) explained 22.31% of the variance among samples, while the second principal coordinate (PCo2) explained 13.97%, with a cumulative explanatory rate of 36.28%. Along the PCo1 axis, samples FPR, FPF, FPL, and PS overlapped, indicating minimal differences and high similarity; samples PF, PL, and PR also overlapped, suggesting small differences and high similarity. Notably, samples FPR, FPL, and FPS were distinctly separated from PF, PL, and PR, reflecting structural differences in endophytic fungi communities between potted and non-potted *C. pareira*. Along the PCo2 axis, FPS was distant from the other seven samples, highlighting distinct diversity in its endophytic fungi. The hierarchical clustering tree in Figure 3b further corroborated these findings, aligning with the patterns observed in the PCoA analysis.

#### 3.1.3. Analysis of Endophytic Fungi Species Composition

Analysis of relative abundance bar charts at the phylum level (Figure 4a) revealed a total of 9 known phyla (*Ascomycota, Basidiomycota, Mortierellomycota, Glomeromycota, Chytridiomycota, Rozellomycota, Olpidiomycota, Mucoromycota*, and *Kickxellomycota*) and one unknown phylum in *C. pareira* across both habitats. Except for samples FPR and PS, where phylum Basidiomycota dominated, phylum Ascomycota was the dominant phylum in all other samples. In potted *C. pareira*, the relative abundance of dominant phyla in roots, stems, leaves, and flowers was 85.23%, 52.35%, 92.77%, and 80.76%, respectively; while in non-potted *C. pareira*, the corresponding values were 76.04%, 63.51%, 66.45%, and 82.83% (Appendix A). At the genus level (Figure 4b), 8 known genera (*Phylloporia*, *Erysiphe*, *Cladosporium*, *Saccharomyces*, *Fusarium*, *Mortierella*, *Thelebolus,* and *Campylospora*) and three unknown genera were identified across both habitats. In potted roots, genera *Campylospora* (18.62%) and *Fusarium* (10.17%) showed higher proportions, while non-potted roots were dominated by *Phylloporia* (74.57%) and *Thelebolus* (3.96%). Potted stems exhibited higher proportions of *Phylloporia* (45.87%) and *Saccharomyces* (8.62%), whereas non-potted stems were dominated by *Mortierella* (14.12%) and *Cladosporium* (2.28%). In potted leaves, *Erysiphe* (54.60%) and *Cladosporium* (6.04%) were predominant, while non-potted leaves showed higher proportions of *Thelebolus* (9.42%) and *Cladosporium* (3.00%). Potted flowers were dominated by *Cladosporium* (16.82%) and *Erysiphe* (7.26%), whereas non-potted flowers featured *Saccharomyces* (15.08%) and *Cladosporium* (8.03%) as predominant genera (Appendix A). In summary, these findings indicate compositional divergence in endophytic fungi between potted and non-potted *C. pareira.* The differences in the distribution of endophytic fungi in potted and non-potted *C. pareira* may be due to environmental factors such as temperature and humidity, and could also be influenced by characteristics such as the health status of the host plant.

### 3.2. Analysis of the Endophytic Fungi Community Structure of C. Pareira Based on the Traditional Culture Methods

Using cultivation-based methods, a total of 69 fungi strains were isolated from the root, stem, leaf, and flower of both potted and non-potted *C. pareira*. Specifically, 32 strains were isolated from potted plants and 37 strains from non-potted plants. The sequences of these strains were subjected to BLAST alignment, and phylogenetic trees were constructed using the neighbor-joining method to identify the most closely related species. The strain sequences were deposited in GenBank via NCBI, and their accession numbers are listed in Table 1, Figure 5, and Appendix A. Appendix A shows the results of partial endophytic fungi isolated from *C. pareira* after 6 days of cultivation on PDA medium.

The combined analysis at the family level and α-glucosidase inhibitory activity (Table 1 and Appendix A, Figure 4 and Appendix A) reveals that 13 strains from 4 families exhibit varying degrees of inhibitory activity. Among them, 8 strains from *Nectriaceae* show inhibitory activity; 3 strains from *Didymellaceae* show inhibitory activity; 1 strain from *Chaetomiaceae* shows inhibitory activity; and 1 strain from *Cladosporiaceae* shows inhibitory activity. The combined analysis at the genus level and glycosidase inhibitory activity shows that 8 strains from 2 genera exhibit varying degrees of inhibitory activity. Among them, 7 strains from *Fusarium* show inhibitory activity, with more than 80% of the strains exhibiting this activity in 3 strains; 1 strain from *Cladosporium* shows inhibitory activity. In summary, *Fusarium* in the *Nectriaceae* family is the dominant genus with glycosidase inhibitory activity.

At the class level, the 32 endophytic fungi strains from potted *C. pareira* were classified into four classes: *Sordariomycetes* (68.75%), *Dothideomycetes* (18.75%), *Eurotiomycetes* (9.38%), and *Agaricomycetes* (3.13%). At the genus level, a total of 16 genera were identified, with 15 genera belonging to Ascomycota and 1 genus to Basidiomycota, and the most abundant genera were *Colletotrichum*, *Fusarium*, *Diaporthe*, and *Aspergillus*. From the root, stem, leaf, and flower tissues, 10, 10, 8, and 4 strains were isolated and identified, respectively. Species richness analysis revealed 8 species (root), 6 species (stem), 3 species (leaf), and 4 species (flower). All 37 endophytic fungi strains from non-potted *C. pareira* belonged to Ascomycota. At the class level, they were classified into three classes: *Sordariomycetes* (67.57%), *Dothideomycetes* (21.62%), and *Eurotiomycetes* (10.81%). At the genus level, 22 genera were identified, with the most abundant being *Colletotrichum*, *Fusarium*, *Diaporthe*, *Xylaria*, *Aspergillus*, *Penicillium*, and *Pestalotiopsis*. From the root, stem, leaf, and flower tissues, 10, 11, 11, and 5 strains were isolated and identified, respectively. Species richness analysis showed 8 species (root), 10 species (stem), 8 species (leaf), and 3 species (flower) (Appendix A).

### 3.3. Analysis of C. Pareira Endophytic Fungi Diversity Based on HTS and Traditional Culture Methods

Using the HTS method, a total of 1607 endophytic fungi species were identified in *C. pareira* from two habitats. In potted *C. pareira*, 677 species were found, spanning 13 phyla, 40 classes, 88 orders, 212 families, and 461 genera, with the dominant genus *Erysiphe* accounting for 15.51% (excluding unidentified genera). In non-potted *C. pareira*, 1154 species were identified, classified into 16 phyla, 47 classes, 103 orders, 259 families, and 680 genera, with the dominant genus *Phylloporia* representing 18.64% (excluding unidentified genera). Using traditional culture-based methods, 32 strains of endophytic fungi were isolated and identified from potted *C. pareira*, belonging to 2 phyla, 4 classes, 8 orders, 12 families, and 16 genera, while 37 strains from non-potted *C. pareira* fell into 1 phylum, 3 classes, 10 order, 20 families, and 22 genera, with the dominant genus *Colletotrichum* accounting for 37.5% and 18.9%, respectively. 9 endophytic fungi strains (e.g., *Penicillium*, *Fusarium*, *Aspergillus*) were shared between *C. pareira* from two habitats. The potted *C. pareira* has 7 unique strains (e.g., *Lasiodiplodia*, *Camarosporium*, *Ilyonectria*), accounting for 21.88%; the non-potted *C. pareira* has 13 unique strains (e.g., *Triangularia*, *Phaeosphaeriopsis*, *Scedosporium*), accounting for 5.14%. Overall analysis shows that both HTS and traditional culture methods detected common genera (e.g., *Cladosporium*, *Fusarium*, *Aspergillus*) in various tissues of *C. pareira*. Simultaneously, each method also identified unique genera (HTS revealed genera such as *Mortierella* and *Aaosphaeria*; traditional culture detected genera including *Triangularia*, *Lasiodiplodia*, and *Humicola*).

### 3.4. α-Glucosidase Inhibitory Activity of Isolated Endophytic Fungi

The results of in vitro α-glucosidase inhibitory activity assays on fermentation products of endophytic fungi from potted and non-potted *C. pareira* are presented in Table 1. The proportions of potted and non-potted endophytic fungi exhibiting α-glucosidase inhibition rates exceeding 50% were 43.75% and 75.68%, respectively. Strains *Aspergillus niger* PXR2, *Fusarium solani* PXR5, and *Pestalotiopsis* sp. PXR10 from potted *C. pareira*, as well as strains *F. solani* FPXR2, *F. nematophilum* FPXR3, *Xylaria apoda* FPXL1, *Nigrospora sphaerica* FPXL3, *Phaeosphaeriopsis musae* FPXL5, *Corynespora cassiicola* FPXL6, *Cladosporium tenuissimum* FPXL10, and *Diaporthe* sp. FPXF4 from non-potted *C. pareira*, demonstrated notable α-glucosidase inhibitory activity with inhibition rates surpassing 80%.

## 4. Discussion

*C. pareira* has been widely used in traditional medicine since ancient times, with various therapeutic effects such as treating ulcers, rheumatism, fever, asthma, cholera, diarrhea, and rabies [43]. Despite its high medicinal value, *C. pareira* is primarily distributed in tropical and subtropical regions, with scarce wild resources and underdeveloped artificial cultivation, and existing research on *C. pareira* has focused predominantly on its chemical constituents and pharmacological activities. Based on the characteristics of endophytic fungi and their metabolites, such as environmental compatibility, non-toxicity, controllable cultivation conditions, rapid growth and reproduction, suitability for large-scale fermentation, and low production costs, endophytic fungi from medicinal plants can serve as one of the new pathways for obtaining bioactive natural products. Moreover, the active metabolites discovered through this approach hold broad application potential in agriculture, industry, and biomedicine [44]. Furthermore, these discoveries advance the sustainable utilization of medicinal plant resources, particularly for species with high medicinal value but slow growth rates and limited wild populations (e.g., expensive, rare plants). By isolating functionally equivalent compounds directly from endophytic fungi, this approach offers a potential solution to resource scarcity. The environment is one of the key factors in the plant growth process, and endophytic fungi in the host are an important component. Therefore, the distribution and types of endophytic fungi are influenced to varying degrees by the environment. In this study, we focused on the endophytic fungi in various tissues of *C. pareira* from different habitats and investigated their diversity in potted and non-potted *C. pareira* using HTS and traditional cultivation methods.

Differences in environmental factors such as soil type and surrounding vegetation may influence microbial population distribution and community structure [45,46]. For instance, Suryanarayanand et al. proposed that the community composition of endophytic fungi in the same plant species exhibits specificity due to geographical variations [47]. Similarly, oak communities within the same region demonstrate significantly higher similarity compared to those from different regions [48]. The richness and diversity of endophytic fungi in *Huperzia serrata* are influenced by tissue type and ecological region, with communities exhibiting species specificity, ecological specificity, and tissue specificity [49]. The findings of this study reveal that both analytical approaches indicate marked differences in the diversity of endophytic fungi (including quantity, community composition, and richness) in *C. pareira* between pot-cultivated and non-pot-cultivated environments. The endophytic fungi diversity in potted *C. pareira* appears relatively homogeneous. This is speculated to be due to the looser soil structure and higher mineral element content in non-pot-cultivated environments. The distribution of endophytic fungi exhibits tissue specificity. Mishra et al. discovered that the richness of endophytic fungi in the medicinal plant *Tinospora sagittata* was closely associated with its tissue sites [50]. Similarly, Chutulo et al. observed significant variations in the richness and diversity of endophytic fungi among the roots, leaves, stems, bark, and branches of the medicinal plant *Azadirachta indica* [51]. This indicates that the tissues of medicinal plants also influence the diversity and richness of their endophytic fungi communities. Furthermore, this study revealed significant differences in the diversity of endophytic fungi in the roots of non-potted *C. pareira* compared to the other three tissues, while the richness of endophytic fungi in the stems also showed significant differences relative to the other three tissues.

Using metagenomic HTS technology to analyze the microbial community structure is currently one of the most important and rapid methods for understanding the structure and function of microbial communities [52]. This technology not only enables direct detection of endophytic fungi within host plant tissues but also provides comprehensive, intuitive, and systematic analysis of plant endophytic fungi through its large sequencing volume and high depth. For example, some researchers have already utilized HTS to investigate the diversity of endophytic fungi communities in medicinal plants such as *Huperzia serrata* [3], *Coptis chinensis* [4], and *Artemisia argyi* [53], obtaining extensive information on endophytic fungi taxa. In contrast, traditional microbial research methods may underestimate the composition and diversity of microbial communities in host plants due to various limitations [54]. However, strains isolated and identified through conventional culture methods can enrich endophytic fungi resource libraries, providing materials for subsequent bioactive studies, while discovered bioactive compounds can demonstrate practical value. Therefore, this study comprehensively employed both HTS and traditional culture methods to investigate endophytic fungi diversity in *C. pareira*. A total of 1607 fungi species were identified from *C. pareira* using HTS, while 69 strains were isolated and identified by traditional culture methods. As mentioned earlier, the number of strains detected by HTS far exceeds that of traditional culture methods, indicating that *C. pareira* is rich in endophytic fungi resources and that comprehensive analysis can be carried out using relevant data. Nevertheless, there are still differences between the two methods in terms of strain composition, such as dominant genera, the types and quantities of unique strains. This demonstrates that HTS and traditional culture methods play complementary and indispensable roles in studying endophytic fungi diversity in *C. pareira*.

α-glucosidase is a critical target enzyme identified in the treatment of type II diabetes. Compared to other oral hypoglycemic drugs, AGIs have become the preferred therapeutic option due to their localized action in the intestinal tract, mild effects, and minimal side effects [37]. However, the currently available AGIs are limited in variety and may cause gastrointestinal and hepatic damage. Therefore, it is of great significance to explore and discover resources for highly effective and low-toxicity AGIs. The metabolites of endophytic fungi in medicinal plants have garnered substantial attention due to their potential α-glucosidase inhibitory activity. Researchers have already discovered metabolites with α-glucosidase inhibitory effects from endophytic fungi of medicinal plants such as *Acanthus ilicifolius* [55], *Costus speciosus* [56], *Paeonia delavayi* [57], and *Simarouba glauca* DC. [58]. In this in vitro α-glucosidase inhibitory activity experiment, a total of 11 endophytic fungi strains demonstrated strong inhibitory activity. Among them, 3 strains were isolated from potted *C. pareira* (PXR2 with 91.86% inhibition rate, PXR5 with 89.67%, and PXR10 with 87.87%), while 8 strains were identified from non-potted *C. pareira* (FPXR2 with 82.08%, FPXR3 with 89.28%, FPXL1 with 86.93%, FPXL3 with 84.04%, FPXL5 with 88.26%, FPXL6 with 86.85%, FPXL10 with 91.39%, and FPXL4 with 94.68% inhibition rates). Notably, strain FPXF4 exhibited inhibitory activity comparable to the positive control, Acarbose. The analysis of the relationship between strain classification and α-glucosidase inhibitory activity levels shows that there are 3 strains of *Fusarium* species, 1 strain of *Cladosporium*, 1 strain of *Corynespora*, 1 strain of *Phaeosphaeriopsis*, 1 strain of *Nigrospora*, 1 strain of *Xylaria*, 1 strain of *Pestalotiopsis*, and 1 strain of *Pestalotiopsis*. The proportion of *Fusarium* species is 27.3%, as shown in Table 1. The results indicate that the endophytic fungi of *C. pareira* from different habitats exhibit diversity in glycosidase inhibition activity, with *Fusarium* species being the dominant genus showing high inhibitory activity. These fungi have potential value in exploring precursor compounds with activity and could be further investigated as a resource for bioactive compounds.

This study still has some areas that need improvement in the overall experimental design, and these limitations may impact the results of the research. For example, in terms of habitat, the factors influencing the diversity of endophytic fungi in *C. pareira* are complex and varied. However, during the experiment, there was a lack of monitoring of temperature and humidity levels in the two habitats, and the study did not delve into the impact of ecosystems and plant sustainability. Additionally, this paper primarily presents the distribution characteristics of OTU diversity, focusing on analyzing the microbial community composition patterns, but it lacks an in-depth discussion of the factors influencing the distribution differences in endophytic fungi in *C. pareira* across different habitats and biological hypotheses regarding the selection of dominant groups. The traditional culture methods used in the isolation of endophytic fungi may not capture all the endophytes, especially those that are difficult to cultivate or are dependent on specific hosts. In future studies, it is necessary to comprehensively consider environmental factors affecting the growth of host plants in the experimental design, including climatic factors, soil factors, biological factors, and anthropogenic factors. The factors affecting the diversity of endophytic fungi will be combined with environmental factors or functional gene analysis to further validate the driving factors of dominant groups, ensuring the systematic nature of the experiment. Traditional culture-based methods for analyzing the diversity of endophytic fungi can involve multiple isolations and purifications from the same sample or increasing the number of isolating tissues to reduce human error and obtain as many strains as possible.

This research mainly focuses on the diversity of endophytic fungi in the above-ground parts of *C. pareira* from two different habitats. Future research can explore aspects such as the ecological functions of microorganisms in the above-ground and below-ground parts, the interaction of metabolites with host plants, and more, rather than being limited to the species identification of endophytic fungi. By fully utilizing high-throughput genomics, metabolomics, and transcriptomics, the functional study of endophytic fungi is gradually shifting from basic research to applied research, exploring their potential applications in agriculture, medicine, and environmental protection. For example, endophytic fungi may play an important role in plant disease resistance, nutrient absorption, and the degradation of environmental pollutants. Future studies will focus on discovering commercially valuable endophytic fungal species and applying them in fields such as green agriculture and natural drug development.

## 5. Conclusions

This study utilized HTS to detect a total of 9587 OTUs from the root, stem, leaf, and flower tissues of both potted and non-potted *C*. *pareira*. These OTUs were classified into 18 phylum, 60 classes, 139 orders, 324 families, 851 genera, and 1607 species, with genus *Phylloporia* sp. being the dominant taxon, accounting for 30.11% of the total (excluding unidentified genera). Through traditional cultivation methods, 69 endophytic fungi strains were isolated and identified, among which the genus *Colletotrichum* sp. was dominant (27.54% abundance). Notably, 11 strains exhibited strong α-glucosidase inhibitory activity. These results provide a data foundation for further research on *C. pareira* as a medicinal resource and hold significant implications for achieving its comprehensive development and sustainable utilization.

## Figures and Tables

**Figure 1 jof-11-00615-f001:**
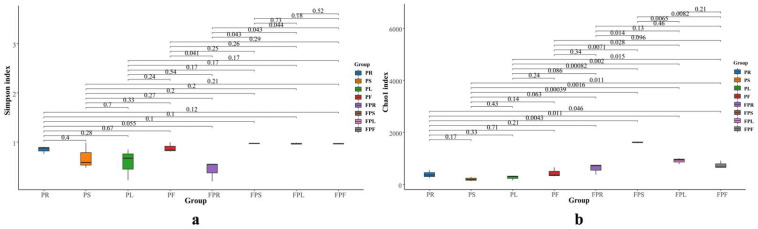
Simpson index of endophytic fungi in different tissues of potted and non-potted *C. pareira* (**a**), chao1 index of endophytic fungi in different tissues of potted and non-potted *C. pareira* (**b**); PR: potted *C. pareira* root, PS: potted *C. pareira* stem, PL: potted *C. pareira* leaf, PF: potted *C. pareira* flower, FPR: non-potted *C. pareira* root, FPS: non-potted *C. pareira* stem, FPL: non-potted *C. pareira* leaf, FPF: non-potted *C. pareira* flower.

**Figure 2 jof-11-00615-f002:**
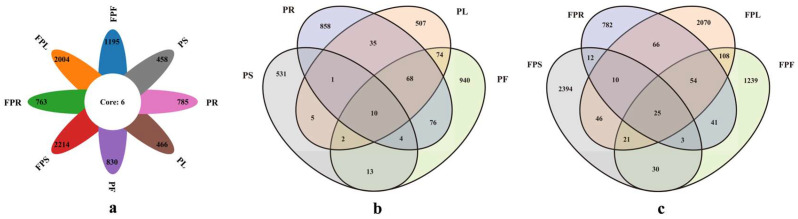
The Venn diagram illustrate the endophytic fungi OTUs in different tissues of potted and non-potted *C. pareira* (**a**), the OTUs in tissue samples of potted *C. pareira* (**b**), the OTUs in tissue samples of non-potted *C. pareira* (**c**); PR: potted *C. pareira* root, PS: potted *C. pareira* stem, PL: potted *C. pareira* leaf, PF: potted *C. pareira* flower, FPR: non-potted *C. pareira* root, FPS: non-potted *C. pareira* stem, FPL: non-potted *C. pareira* leaf, FPF: non-potted *C. pareira* flower.

**Figure 3 jof-11-00615-f003:**
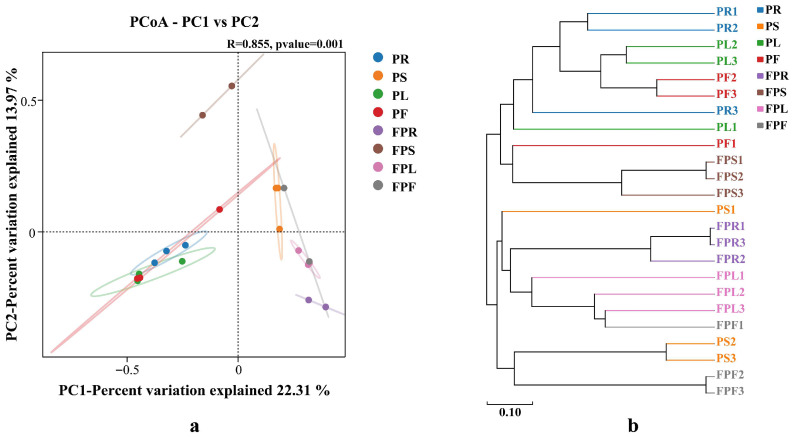
Principal coordinates analysis based on Bray–Curtis (**a**). Cluster analysis based on Bray–Curtis (**b**); PR: potted *C. pareira* root, PS: potted *C. pareira* stem, PL: potted *C. pareira* leaf, PF: potted *C. pareira* flower, FPR: non-potted *C. pareira* root, FPS: non-potted *C. pareira* stem, FPL: non-potted *C. pareira* leaf, FPF: non-potted *C. pareira* flower. The numbers 1, 2, and 3 in the names are used to indicate that the same tissue sample is repeated three times.

**Figure 4 jof-11-00615-f004:**
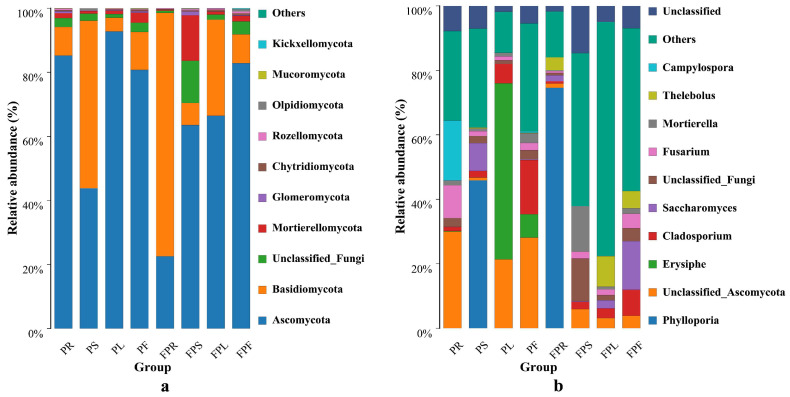
Fungi community composition at the phylum level (**a**). Fungi community composition at the family level (**b**); PR: potted *C. pareira* root, PS: potted *C. pareira* stem, PL: potted *C. pareira* leaf, PF: potted *C. pareira* flower, FPR: non-potted *C. pareira* root, FPS: non-potted *C. pareira* stem, FPL: non-potted *C. pareira* leaf, FPF: non-potted *C. pareira* flower.

**Figure 5 jof-11-00615-f005:**
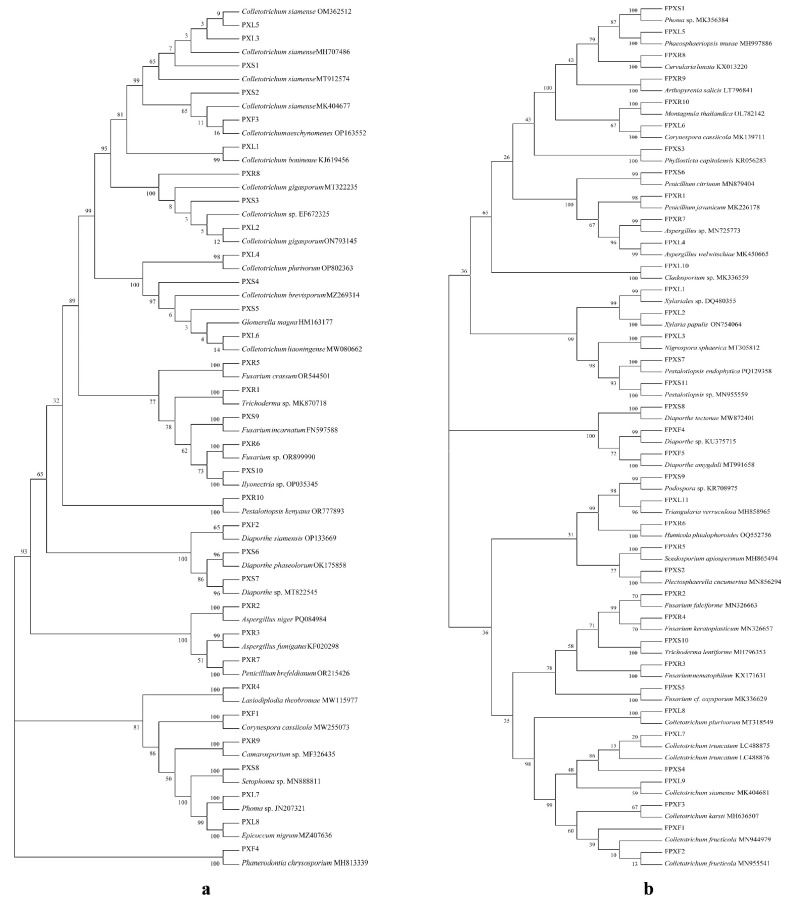
Phyogenetic analysis of rDNA-ITS sequences of endophytic fungi separated from potted *C. pareira* (**a**); Phyogenetic analysis of rDNA-ITS sequences of endophytic fungi separated from non-potted *C. pareira* (**b**). (Phylogenetic trees were constructed using the Neighbor-Joining method in MEGA 7.0 software, based on the Kimura 2-parameter distance. The trees were tested using the bootstrap method with 1000 iterations.).

**Table 1 jof-11-00615-t001:** Endophytic fungi isolated from *C. pareira* using traditional culture methods and their α-glucosidase inhibitory activity.

Source	Strains	Closest Relatives in NCBI (Accession No.)	Max. Identity %	GenBank No.	Antiglucosidase Inhibitory % Activity (%)
Potted root	PXR1	*Trichoderma afroharzianum* (MN644718)	100.00	PQ345005	9.55 ± 6.17
	PXR2	*Aspergillus niger* (OP861489)	100.00	PQ345006	91.86 ± 0.59 ^▲^
	PXR3	*Aspergillus fumigatus* (OP103933)	100.00	PQ345007	53.91 ± 2.58
	PXR4	*Lasiodiplodia brasiliense* (KC484814)	100.00	PQ345008	45.93 ± 3.99
	PXR5	*Fusarium solani* (OP117297)	100.00	PQ345009	89.67 ± 3.05 ^▲^
	PXR6	*Fusarium nematophilum* (LC317606)	100.00	PQ345010	51.56 ± 0.36
	PXR7	*Penicillium brefeldianum* (MH864250)	100.00	PQ345011	/
	PXR8	*Colletotrichum* sp. (OK030894)	100.00	PQ345012	36.62 ± 3.69
	PXR9	*Camarosporium* sp. (KJ780771)	100.00	PQ345013	26.13 ± 9.13
	PXR10	*Pestalotiopsis* sp. (HE608797)	100.00	PQ345014	87.87 ± 2.51 ^▲^
Potted stem	PXS1	*Colletotrichum siamense* (MN519187)	100.00	PQ345015	43.11 ± 2.59
	PXS2	*Colletotrichum aenigma* (OM663733)	100.00	PQ345016	31.92 ± 5.54
	PXS3	*Colletotrichum* sp. (OK030894)	100.00	PQ345017	69.25 ± 0.62
	PXS4	*Colletotrichum liaoningense* (MW349986)	100.00	PQ345018	32.71 ± 5.65
	PXS5	*Glomerella magna* (HM163187)	99.45	PQ345019	51.96 ± 5.87
	PXS6	*Diaporthe rosae* (OQ793617)	100.00	PQ345020	76.45 ± 1.56
	PXS7	*Diaporthe phaseolorum* (MK448274)	100.00	PQ345021	64.40 ± 4.03
	PXS8	*Setophoma* sp. (OP392554)	99.62	PQ345022	37.25 ± 6.48
	PXS9	*Fusarium* sp. (KP006638)	100.00	PQ345023	67.06 ± 7.46
	PXS10	*Ilyonectria* sp. (MZ374709)	100.00	PQ345024	20.89 ± 6.35
Potted leaf	PXL1	*Colletotrichum karsti* (LC494365)	100.00	PQ345025	23.32 ± 8.92
	PXL2	*Colletotrichum gigasporum* (OM397123)	100.00	PQ345026	6.26 ± 5.73
	PXL3	*Colletotrichum* sp.(MT241882)	100.00	PQ345027	61.35 ± 2.70
	PXL4	*Colletotrichum plurivorum* (MT318545)	100.00	PQ345028	0.94 ± 13.62
	PXL5	*Colletotrichum siamense* (OL966214)	100.00	PQ345029	43.82 ± 2.90
	PXL6	*Colletotrichum brevisporum* (MZ269314)	100.00	PQ345030	54.38 ± 1.56
	PXL7	*Phoma* sp. (KX216732)	100.00	PQ345031	18.23 ± 1.11
	PXL8	*Epicoccum nigrum* (MW081246)	100.00	PQ345032	43.51 ± 0.98
Potted flower	PXF1	*Corynespora cassiicola* (OM802573)	100.00	PQ345033	35.05 ± 1.91
	PXF2	*Diaporthe* sp. (MT822641)	100.00	PQ345034	76.13 ± 3.74
	PXF3	*Colletotrichum* sp. (MW786244)	100.00	PQ345035	34.98 ± 1.64
	PXF4	*Phanerodontia chrysosporium* (MW38799)	100.00	PQ345036	66.67 ± 3.61
non-potted root	FPXR1	*Penicillium javanicum* (MK450698)	100.00	PQ344968	29.11 ± 5.52
	FPXR2	*Fusarium solani* (MK336609)	100.00	PQ344969	82.08 ± 5.33 ^▲^
	FPXR3	*Fusarium nematophilum* (MN540302)	100.00	PQ344970	89.28 ± 3.56 ^▲^
	FPXR4	*Fusarium keratoplasticum* (MN326661)	100.00	PQ344971	26.45 ± 7.68
	FPXR5	*Scedosporium apiospermum* (MN177623)	100.00	PQ344972	74.57 ± 2.53
	FPXR6	*Humicola* sp. (KY069223)	100.00	PQ344973	52.66 ± 6.03
	FPXR7	*Aspergillus terreus* (MN592939)	100.00	PQ344974	43.97 ± 5.17
	FPXR8	*Curvularia lunata* (MG649266)	100.00	PQ344975	20.03 ± 1.43
	FPXR9	*Arthopyrenia salicis* (LT796900)	100.00	PQ344976	79.81 ± 1.69
	FPXR10	*Montagnula graminicola* (PP886946)	100.00	PQ344977	79.34 ± 1.64
	FPXS1	*Phoma* sp. (MT251173)	100.00	PQ344978	79.26 ± 3.53
	FPXS2	*Plectosphaerella cucumerina* (MN856305)	100.00	PQ344979	60.56 ± 5.31
non-potted stem	FPXS3	*Phyllosticta rhizophorae* (MT360030)	100.00	PQ344980	70.58 ± 2.23
	FPXS4	*Colletotrichum* sp. (MT476751)	100.00	PQ344981	16.35 ± 9.22
	FPXS5	*Fusarium oxysporum* (MK752409)	100.00	PQ344982	51.25 ± 4.84
	FPXS6	*Penicillium citrinum* (LC514694)	100.00	PQ344983	51.80 ± 0.89
	FPXS7	*Pestalotiopsis* sp. (JF773654)	99.82	PQ344984	69.25 ± 3.69
	FPXS8	*Diaporthe tectonae* (PP060674)	100.00	PQ344985	62.83 ± 3.26
	FPXS9	*Podospora* sp. (OQ413591)	99.79	PQ344986	11.89 ± 11.07
	FPXS10	*Trichoderma harzianum* (MT341774)	100.00	PQ344987	53.05 ± 5.74
	FPXS11	*Pestalotiopsis mangiferae* (KM510409)	100.00	PQ344988	77.62 ± 1.79
non-potted leaf	FPXL1	*Xylaria apoda* (MZ423071)	99.49	PQ344989	86.93 ± 2.44 ^▲^
	FPXL2	*Xylaria* sp. (OR122882)	99.82	PQ344990	34.90 ± 2.28
	FPXL3	*Nigrospora sphaerica* (MT305813)	100.00	PQ344991	84.04 ± 3.64 ^▲^
	FPXL4	*Aspergillus welwitschiae* (MK450668)	100.00	PQ344992	52.19 ± 2.40
	FPXL5	*Phaeosphaeriopsis musae* (MT071749)	100.00	PQ344993	88.26 ± 3.73 ^▲^
	FPXL6	*Corynespora cassiicola* (FJ852574)	100.00	PQ344994	86.85 ± 5.92 ^▲^
	FPXL7	*Colletotrichum truncatum* (MK298334)	100.00	PQ344995	78.25 ± 7.81
	FPXL8	*Colletotrichum cliviicola* (MT351124)	99.80	PQ344996	73.24 ± 3.41
	FPXL9	*Colletotrichum siamense* (MK404694)	100.00	PQ344997	44.99 ± 0.36
	FPXL10	*Cladosporium tenuissimum* (MN700643)	100.00	PQ344998	91.39 ± 0.68 ^▲^
	FPXL11	*Triangularia* sp. (OR825378)	100.00	PQ344999	76.21 ± 0.95
non-potted flower	FPXF1	*Colletotrichum alienum* (MK336495)	100.00	PQ345000	59.23 ± 1.18
	FPXF2	*Colletotrichum* sp. (MK351442)	100.00	PQ345001	60.72 ± 0.68
	FPXF3	*Colletotrichum karsti* (MT319073)	100.00	PQ345002	54.38 ± 4.87
	FPXF4	*Diaporthe* sp. (KC507214)	99.59	PQ345003	94.68 ±2.94 ^▲^
	FPXF5	*Diaporthe ternstroemia* (NR147523)	99.60	PQ345004	*/*
control	Acarbose	*/*	*/*	*/*	93.43 ± 3.08

Note: Strains PXR1-PXR10 were isolated from roots of potted *C. pareira*; strains PXS1-PXS10 were isolated from stems of potted *C. pareira*; strains PXL1-PXL8 were isolated from leaves of potted *C. pareira*; strains PXF1-PXF4 were isolated from flowers of potted *C. pareira*. Strains FPXR1-FPXR10 were isolated from roots of non-potted *C. pareira*; strains FPXS1-FPXS11 were isolated from stems of non-potted *C. pareira*; strains FPXL1-FPXL11 were isolated from leaves of non-potted *C. pareira*; strains FPXF1-FPXF5 were isolated from flowers of non-potted *C. pareira*. The experiment to test the inhibition of α-glucosidase activity by fungi extracts was repeated three times, with the results expressed as the mean ± SD of three replicates. “/” indicates no result or no such item. “^▲^” indicates the strong activity.

## Data Availability

The sequence data from the endophytic fungi of *Cissampelos pareira* were deposited in the Sequence Read Archive of the NCBI under accession number PRJNA1280751. The ITS rRNA gene sequences of culturable endophytic fungi strains have been deposited in the GenBank database with accession numbers PQ344968-PQ345036.

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
