# Peer review of "Community Structure Diversity of Endophytic Fungi in Cissampelos pareira from Different Habitats and Their α-Glucosidase Inhibitory Activity"

_jof, 2025, doi:10.3390/jof11090615_

Round 1

Reviewer 1 Report

Review of the article ‘Community Structure Diversity of Endophytic Fungi in Cissampelos pareira From Different Habitats and Their α-Glucosidase Inhibitory Activity’

This article analyses the structure and diversity of endophytic fungi in Cissampelos pareira from different habitats, providing a comparative assessment of their inhibitory activity against α-glucosidase. The study is comprehensive, combining high-throughput sequencing methods with classical cultural microbiology techniques. The interdisciplinary nature of the research (combining microbiology, pharmacology and bioinformatics) is evident throughout.

Main shortcomings and comments:

  1. The abstract is too long and should be shorter.
  2. The authors outline the well-known benefits of studying endophytes and offer generalised conclusions regarding the potential of these microorganisms for pharmacological applications. The literature review (page 2, lines 55–61) lacks a detailed analysis of what makes this study unique compared to the numerous other publications on endophyte diversity in medicinal plants and its contribution to the field's development. Recommendation: The innovative aspects should be more clearly identified (e.g. these habitats were studied for the first time, or the comparative approach of ‘potted vs non-potted plants’).
  3. The discussion only cites selected results from similar studies and does not compare the main parameters (diversity, dominant taxa and characteristics of inhibitory strains) with those from previously published studies (see p. 13, lines 407–428).
  4. The authors identified strains with high inhibitory activity, but did not establish a detailed connection between specific taxa (or groups of taxa) and the level of activity. Nor did they perform any correlation analysis between community structure and bioactivity. (p. 14, lines 492–505; Table 1, pp. 9–10).
  5. Although the Chao1 and Simpson indices are used (see Fig. 2), there is no detailed description of the statistical tests performed. P-values are rarely given and there are minimal detailed comparisons between groups.
  6. Habitats are only labelled as 'potted' or 'non-potted', with no information given about the factors that differentiate them, such as soil structure, moisture levels and lighting conditions. The consequences for ecosystem and plant sustainability are hardly discussed either. Recommendation: Describe the environmental conditions of both models in more detail (see p. 13, lines 440–453) or acknowledge the limitations of such data.
  7. Although it is stated that the experiments were repeated three times, it is not always clear how many biological and technical repetitions accompanied the analyses. For some of the biotests, positive and negative controls are not indicated. For example, for inhibition, the use of 'acarbose' is indicated as a control, but the methodology is described briefly (see p. 5, lines 199–210).
  8. Although the visualisation (Figs. 1 and 3) provides an overview of OTU diversity, the text does not attempt to explain the reasons for structural differences or offer biological hypotheses about dominant group selection (see Figs. 5a–b, p. 8, lines 306–332).
  9. The methods for isolating and testing bioactivity are described briefly.

    Additional comments on formatting and structure:

    In the 'Results' section, individual paragraphs are duplicated (e.g. those on unique and common OTUs) — these could be shortened to capture the essence (see lines 260–283).

    It is recommended that 'Limitations of the study' and 'Prospects for further work' be separated and placed at the end of the 'Discussion' section (pp. 14–15, lines 492–516).

    Conclusion:

    The work is of high quality and corresponds to the journal's subject area. The data will be of interest to specialists in microbiology, biotechnology and pharmacognosy. However, the aforementioned shortcomings reduce the interpretability, novelty and practical significance of the article. Once the above comments have been addressed (i.e. the methods have been detailed, the statistical analysis has been expanded, correlations have been established, and limitations and hypotheses have been discussed), the manuscript can be recommended for publication.

    English language level: average; requires editing.

    While the text is essentially understandable, it would benefit significantly from professional editing. To be published in prestigious international journals, we recommend the following:

    1. Have the text thoroughly proofread by a native speaker or professional scientific editor.
    2. Correct any structural or grammatical errors.
    3. Reduce excessive repetition.
    4. Rewrite complex sentences to make them simpler and more natural.
    5. Check the terminology and style against publications in leading microbiology and biotechnology journals.

    Significant language editing is required to achieve a standard consistent with international scientific publications.

References to graphs and tables:

Figure 1 (page 6, lines 227–232): Shows saturation curves for samples — analyse how well rare taxa are captured. Figure 1 reflects saturation curves and has an auxiliary technical function of checking the completeness of sequencing. To improve the informativeness of the main text and the compactness of the data presentation, it is recommended that this figure be moved to the Supplementary Materials section, with a link to it and a brief description of the conclusions remaining in the main text.

Figure 2 (pp. 6–7, lines 234–257): Diversity indices for each group. It is important to add statistics on differences directly to the graphs.

Figure 3 (p. 7, lines 259–283): Venn diagrams. Justification for the highlighted common and unique OTUs should be added.

Figure 5 (p. 8, lines 328–332): Taxonomic comparison by family and genus. It would be useful to compare this with inhibitory activity.

Figure 6: Have you constructed the phylogenetic tree correctly? For example, the Fusarium isolate in the phylogenetic tree in Figure 6 is incorrectly grouped with a Trichoderma representative in one clade, contradicting current ideas about the Hypocreaceae phylogeny and requiring a revision of the tree's structure or annotation.

Table 1 (pp. 9–10, lines 343–352): This summarises strains and % inhibition, but needs structural refinement (e.g. highlight strains with maximum activity using colour or symbols).

Additional materials (page 15, lines 518–526): The authors refer to additional tables and figures, which should be described in the results and discussion section of the text.

Table S1, to which the authors refer in the supplementary materials section, cannot be viewed — either the file is missing or it is damaged.

The authors use terminology related to microbiome analysis repeatedly throughout the manuscript, particularly in the following sections:

The term 'OTU' (Operational Taxonomic Unit) is consistently used to describe sequence clustering (see, for example, p. 6, lines 157–279 and Fig. 3). The term 'ASV' (Amplicon Sequence Variant) does not appear in the main body of the text, in the captions of figures or tables, or in the main text, but it is used in the supplementary materials. The principle used to select the units of analysis (OTU vs. ASV) should be clearly defined and applied consistently, as modern journal standards require a distinction to be made between OTUs and ASVs. This distinction directly affects the reproducibility and comparability of results.

The general layout of the illustrative material:

Some figures lack clear axis labels and the captions and labels use font sizes that are too small, which does not comply with scientific visualisation standards. In accordance with the journal's editorial requirements, all captions and labels must be legible when enlarged to the standard publication size (A4). Most of the figures in the attached PDF file (e.g. figures 2–5) are low resolution and blurred.

Reviewer 2 Report

The manuscript by Yu et al. explores the Endophytic the fungal communities of roots, stems, leaves, and flowers from the traditional antidiabetic medicinal plant Cissampelos pareira, under potted and non-potted conditions using high-throughput sequencing and culture methods. Non-potted plants showed greater fungal richness and diversity, with community composition varying by tissue type. Sixty-nine strains were isolated, of which 11 exhibited strong α-glucosidase inhibition (>80%), indicating significant antidiabetic potential and supporting the sustainable utilization of C. pareira resources.

The manuscript addresses both traditional and high-throughput sequencing approaches. To further strengthen the discussion regarding the nature of the populations identified in the potted model, I would suggest considering the type of soil used for cultivation and its potential impact on the observed results.

-

Author Response

Thank you very much for your comment

Reviewer 3 Report

The revised manuscript provides important data on the diversity and structure of endophytic fungal communities in different organs of the medicinal plant C. pareira. The data are novel, and the study employs both traditional cultivation methods and molecular biology to characterize the communities and provides information on their richness, composition, and alpha-glucosidase inhibitory activity of these fungi located in different organs of the medicinal plant under study. The manuscript is consistent with the journal's scope and is worthy of publication. 

I found some aspects in the current version of this manuscript that could be improved by the authors. These aspects are highlighted throughout the manuscript in the attached PDF file. My recommendation is that, once the authors incorporate these aspects, the manuscript can be published.

Round 2

Reviewer 1 Report

The article can be accepted for publication.

The article can be accepted for publication.